# Systematic analysis of SARS-CoV-2 Omicron subvariants' impact on B and T cell epitopes

**Ruba Al Khalaf, Anna Bernasconi** ⑩ *, **Pietro Pinoli** ⑩

Dipartimento di Elettronica, Informazione e Bioingegneria (DEIB), Politecnico di Milano, Milano, Italia

* anna.bernasconi@polimi.it

## Abstract

### Introduction

Epitopes are specific structures in antigens that are recognized by the immune system. They are widely used in the context of immunology-related applications, such as vaccine development, drug design, and diagnosis / treatment / prevention of disease. The SARS-CoV-2 virus has represented the main point of interest within the viral and genomic surveillance community in the last four years. Its ability to mutate and acquire new characteristics while it reorganizes into new variants has been analyzed from many perspectives. Understanding how epitopes are impacted by mutations that accumulate on the protein level cannot be underrated.

### Methods

With a focus on Omicron-named SARS-CoV-2 lineages, including the last WHO-designated Variants of Interest, we propose a workflow for data retrieval, integration, and analysis pipeline for conducting a database-wide study on the impact of lineages' characterizing mutations on all T cell and B cell linear epitopes collected in the Immune Epitope Database (IEDB) for SARS-CoV-2.

### Results

Our workflow allows us to showcase novel qualitative and quantitative results on 1) coverage of viral proteins by deposited epitopes; 2) distribution of epitopes that are mutated across Omicron variants; 3) distribution of Omicron characterizing mutations across epitopes. Results are discussed based on the type of epitope, the response frequency of the assays, and the sample size. Our proposed workflow can be reproduced at any point in time, given updated variant characterizations and epitopes from IEDB, thereby guaranteeing to observe a quantitative landscape of mutations' impact on demand.

### Conclusion

A big data-driven analysis such as the one provided here can inform the next genomic surveillance policies in combatting SARS-CoV-2 and future epidemic viruses.

**Data Availability Statement:** The analyzed data, code workflow (as a Jupyter Notebook), and supplementary materials are available on https://zenodo.org/doi/10.5281/zenodo.10514577.

**Funding:** This work has been funded by Ministero dell'Università della Ricerca (PRIN PNRR 2022 "SENSIBLE" project, n. P2022CNN2J), Principal Investigator: Anna Bernasconi.

**Competing interests:** The authors have declared that no competing interests exist.

## Introduction

Specific sequences of amino acid residues in a viral protein, called epitopes, can be recognized by antibodies or B/T cell receptors as part of the induction of T cell-dependent cellular immune response [1] or B cell-dependent humoral immune response from the host organism [2]. Studying SARS-CoV-2 epitopes is essential, especially in vaccine design; notably, a great number of epitopes available for the Spike protein of SARS-CoV-2 are used in the design of COVID-19 vaccines [3], following multi-epitopes designs [4]. Within the Immune Epitope Database (IEDB, [5]), epitopes have been deposited for many viral species. The ones initially designed for SARS-CoV-2 were derived from SARS [6]; the first 283 epitopes for the virus were deposited by August 2020, reaching -to date- almost 6K units. Candidates for next-generation COVID-19 vaccines can be revealed by identifying targets of broadly neutralizing antibody responses and immunodominant T cell epitopes [7, 8]. Observing epitope variability has applications in disease monitoring, diagnostic settings, as well as drug design [9]. For both B and T cell epitope-based vaccine design [10, 11], it is also important to study epitopes' conservation with respect to mutations accumulated through evolution by the virus. Notably, a mutation occurring on the specific epitope range may affect the recognition of the epitope.

Starting in late November 2021, the Omicron variant (B.1.1.529)—with its 40 non-synonymous mutations in the Spike protein – captured the international community's attention as a potentially highly critical SARS-CoV-2 variant. On November 26th, 2021, Omicron was labeled a Variant Of Concern (VOC) by the World Health Organization (WHO) and many of its descending lineages were considered VOCs too. On March 15th, 2023 [12] the WHO updated its definitions and criteria for the classification of SARS-CoV-2 variants, respectively of Variants of Interest (VOI) and Variants Under Monitoring (VUM) [13]. According to the statement, Omicron has been considered the most divergent VOC observed thus far. At the time of writing, five lineages descending from Omicron are considered VOIs and five VUMs.

Since its appearance, Omicron viruses have undergone genetic and antigenic changes, giving rise to an increasing array of sublineages. All sublineages share common traits, such as the ability to evade the immunity present in the population and a predilection for infecting the upper respiratory tract rather than the lower respiratory tract [14], distinguishing them from variants of concern that emerged before Omicron. Compared to other SARS-CoV-2 variants, Omicron is considered to have a higher non-synonymous mutation rate [15], a lower disease severity [16], and higher transmission [17, 18], which are all features that might lead a virus to quickly evolve into several sublineages. In the PANGO nomenclature [19], sublineages have been named using the BA alias; four are the sublineages descending from B.1.1.529 that are circulating (i.e., BA.1, BA.2, BA.4, and BA.5). During the pandemic wave happening in the winter of 2021, BA.1 was the dominant lineage; it was then replaced with BA.2 and BA.2.12.1, which were -in turn- replaced by BA.4 and BA.5 during the summer of 2022 [8].

As part of convergent evolution [20], genetic recombination has been observed as a genetic event within the Omicron genome [21, 22]. According to Shiraz and Tripathi [23], there was an extraordinary increase in the emergence of SARS-CoV-2 recombinant lineages during the Omicron waves; also, they noted the enrichment of certain amino acids in the Spike protein of recombinant lineages, which have been reported to confer immune escape from neutralizing antibodies and increase the binding to the host's angiotensin-converting enzyme 2 receptor in some cases.

The emergence and rapid spread of the Omicron variant has highlighted that the COVID-19 pandemic is still ongoing, even if the virus may evolve toward an endemic seasonal upper respiratory tract infection. The importance of vaccination persists as a fundamental element of

the strategy to manage the pandemic, even in the face of variants that can evade immunity [24]. Genomic surveillance, the study of the evolution of a pathogen through the sequencing of its genome, was universally recognized as a first line of defense to combat the pandemic [25].

Along with continuous monitoring of new sequences (performed with tools such as Viru-Surf [26]), their mutations' effects (that we categorized in the CoV2K model [27]), and how they aggregate into new variants [28, 29], it is crucial to observe how these impact epitopes [30]. To this end, we first proposed to study epitopes in the context of genomic surveillance in our EpiSurf tool [31], which allows user-friendly testing of epitope conservancy within selected populations of interest. Specifically, EpiSurf computes the number of mutations on epitopes for a specific population of sequences; these can be visualized through VirusViz [32].

It is important to understand how the currently employed set of epitopes is impacted by currently circulating variants. In this contribution, we select the time in the SARS-CoV-2 pandemic when a new generation of variants—led by the 'Omicron' name—emerged and we produce a database-wide data analysis on how these variants behaved on the epitopes locations. We collected all linear SARS-CoV-2 B cell and T cell epitopes on the different proteins, curated by the Immune Epitope Database and Analysis Resource and information on variants and characterizing mutations from CoVariants.org [33], building up on our preliminary observations made in the Virological.org forum post [34], reported as soon as the first samples of Omicron (B.1.1.529) were made available on public databases (Nov. 30th, 2021). Our big data study allows us to draw conclusions on the quantitative relationship between epitopes and evolving SARS-CoV-2 variability, contributing to informing the next health policy-making strategies.

## Materials and methods

We carried out a complete data science pipeline comprising *Data retrieval* from publicly available databases, *Data integration* for achieving information interoperability, *Data aggregation* to consider epitope-level information, and *Data analysis* applying statistical tests. The data analysis steps have been repeated on a full epitopes dataset and also on a restricted dataset with only high-frequncy epitopes. Fig 1 presents the workflow used in this research. The analysis has been performed in Python (Version 3.9.13), using classical data science libraries, i.e., Pandas (Version 1.4.4) for data extraction and aggregation, Scipy (Version 1.9.1) for statistical analysis, Seaborn (Version 0.11.2) and Matplotlib (Version 3.5.2) for data visualization.

### Data retrieval

**Epitopes datasets.** The epitopes datasets `tcell_full_v3` and `bcell_full_v3` were retrieved as zip files from the Immune Epitope Database and Analysis Resource (IEDB) [5] on August 20th, 2023 from [35]. Note that IEDB only includes candidates with a maximum length of 50 amino acids and a maximum non-peptidic structure of 5K Daltons [36], experimentally tested for binding to an adaptive immune receptor (T cell receptor (TCR), antibody or B cell receptor (BCR), or major histocompatibility complex (MHC)) or with a receptor recognized to be epitope-specific.

The dataset on T cell epitopes contains 504,901 entries, each corresponding to one assay performed to identify an epitope. Such entries are described by 160 manually curated metadata fields (organized into 21 sections); 22,774 of them have been derived from Severe acute respiratory syndrome coronavirus 2 (SARS-CoV-2) genomes. We focus on the 22,752 ones (99.90%) that refer to linear peptides, while only 22 (0.09%) refer to discontinuous peptides. Overall, assays refer to 5,927 unique linear epitopes, which are considered in our analysis. Epitopes are derived from 14 SARS-CoV-2 proteins and are obtained from three host species,

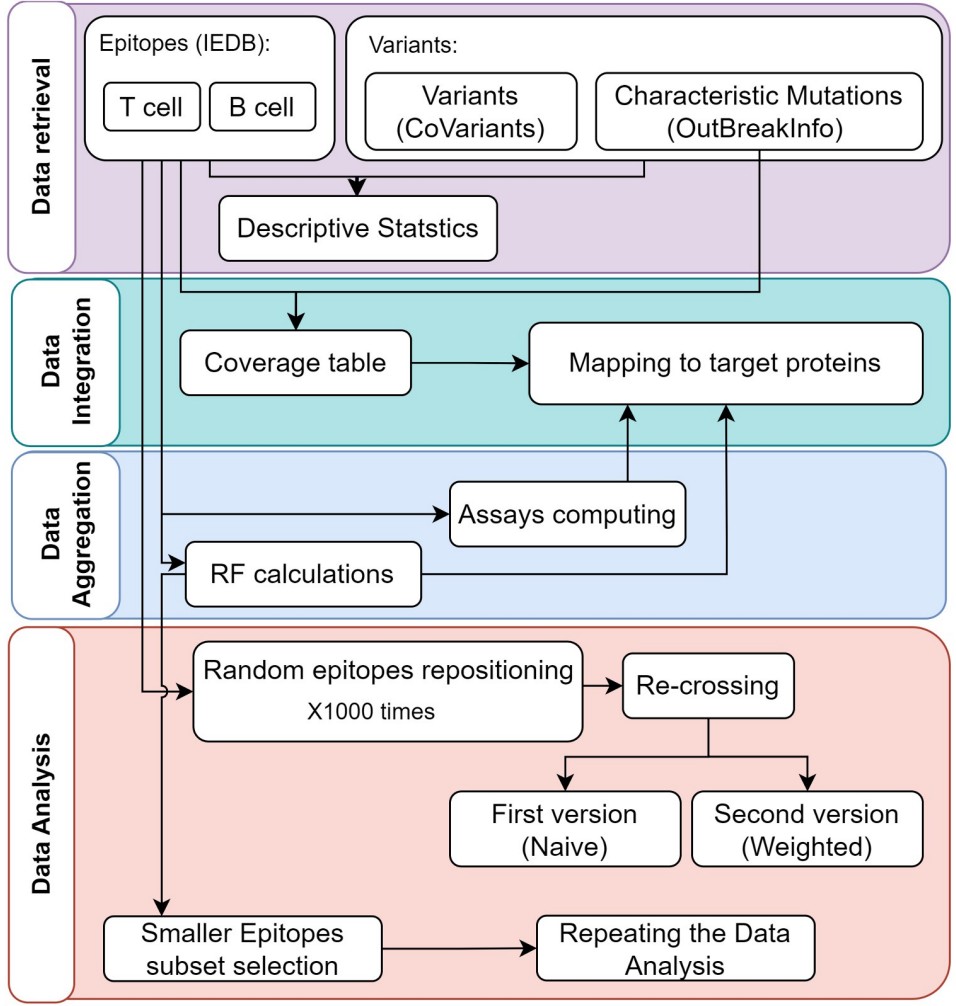

**Fig 1. Analysis workflow, divided into four phases.** Each block corresponds to a code module; arrows indicate the use of input (incoming arrow) calculated in previous steps (outgoing arrow). Note that RF stands for Response Frequency, indicating the number of positively responding subjects over the total number of tested subjects.

i.e., Homo sapiens (human), Mus musculus (mouse), and Macaca mulatta (rhesus macaque). Table 1 shows the counts of T cell (left) and B cell (right) linear epitopes over the SARS-CoV-2 proteins and their average length; the highest amount of linear T cell epitopes (2,794) is derived from the Spike protein, which is to be expected since Spike is an immunogenic protein that mediates host cell entry.

Instead, the dataset on B cell epitopes contains 1,387,901 entries regarding assays, described by 131 manually-curated metadata fields (organized into 11 sections); 90,344 entries are related to the SARS-CoV-2 species. Out of these, 79,364 (87.84%) refer to linear peptides (on which we focus), 10,679 (11.82%) to discontinuous peptides, and 301 (0.33%) to discontinuous multi-chain peptides. Assays refer to 20,388 unique epitopes, which we consider in the analysis. Here, epitopes are derived from 13 SARS-CoV-2 proteins. They are obtained from 15 host species, of which the three most presented ones are Homo sapiens (human), Macaca mulatta (rhesus macaque), and Mus musculus (mouse). In Table 1, we observe that the highest number of linear B cell epitopes is derived from the polyprotein ORF1ab (13,326 epitopes)—not

**Table 1. Overview of quantitative charactristics of T cell and B cell linear epitopes considered in the input datasets.**

| Protein | | T cell linear epitopes | | | | B cell linear epitopes | | | |
|---|---|---|---|---|---|---|---|---|---|
| Name | Length | Count | Min length | Max length | Avg length | Count | Min. length | Max length | Avg length |
| S | 1273 | 2794 | 7 | 43 | 15.17 | 4323 | 4 | 48 | 16.05 |
| N | 419 | 603 | 7 | 38 | 13.95 | 1101 | 4 | 47 | 16.14 |
| M | 222 | 326 | 8 | 41 | 13.90 | 397 | 10 | 32 | 15.89 |
| E | 75 | 86 | 8 | 21 | 12.80 | 123 | 10 | 23 | 15.74 |
| ORF1ab | 7079 | 1724 | 8 | 29 | 11.22 | 13326 | 11 | 48 | 15.48 |
| ORF3a | 275 | 171 | 5 | 38 | 13.63 | 466 | 7 | 34 | 15.32 |
| ORF6 | 61 | 29 | 9 | 31 | 13.86 | 87 | 11 | 16 | 15.14 |
| ORF7a | 121 | 69 | 8 | 35 | 13.20 | 206 | 11 | 17 | 15.04 |
| ORF7b | 43 | 9 | 8 | 25 | 11.44 | 35 | 12 | 16 | 15.63 |
| ORF8 | 121 | 72 | 8 | 37 | 13.86 | 248 | 11 | 22 | 14.83 |
| ORF9b | 97 | 18 | 8 | 20 | 13.94 | 1 | 42 | 42 | 42.00 |
| ORF9c | 73 | 10 | 9 | 20 | 15.60 | 0 | 0 | 0 | 0 |
| ORF10 | 38 | 14 | 9 | 22 | 12.21 | 68 | 12 | 24 | 14.44 |

surprisingly, as ORF1ab accounts for two-thirds of the genome. The ORF1ab of SARS-CoV-2 is considered a potential drug target because it encodes a large polyprotein that is subsequently cleaved into various non-structural proteins (NSPs) essential for viral replication.

In the datasets, each entry represents one assay performed to analyze or devise an epitope: the same epitope appears in many assays (i.e., rows). Metadata is organized into sections: the "Reference" section contains the metadata of the journal publications where the related assays have been studied and reported; it includes fields such as IRI, Type, PMID, submission ID, and Authors. The "Epitope" section contains the metadata of the epitopes, including fields such as the IEDB IRI, Object Type, Name, Starting Position, and Ending Position. The "Related Object" section describes the epitopes' structure (i.e., analog, mimotope, neoepitope, or another structure). The "Host" section contains the metadata of the organism whose T/B cell response is being measured, including Name, IRI, Geolocation, etc. The "1st in vivo process" section contains the metadata of the in vivo process by means of which the organism generating T/B cells was exposed to a relevant immunogen in vivo; it includes fields such as Process Type and Disease. The "Assay" section contains the metadata of the experimental assay including fields Method, Response measured, Units, Qualitative Measurement, Measurement Inequality, Quantitative measurement, Number of Subjects Tested, Number of Subjects Positive, and Response Frequency (%).

In this work, we focus on metadata derived from the "Reference", "Epitope" and "Assay" sections. In addition to metadata *as is* we extend the dataset by calculating the number of positive and negative assays and the Response Frequency aggregating results by epitope; this serves the purpose of exploring how often each protein region has been studied in different immune assays and in how many assays the immune response was positive or negative. Note that, due to the heterogeneity of samples and the complexity of the immune response, the response varies among studies and assays.

**Variants information.** We selected all Omicron subvariants indicated on CoVariants.org [33] as of August 29th, 2023. The 14 variants are shown in Table 2, where BA.2 has the highest number of collected sequences from GISAID [37]. Specifically, we restrict to all the clades that descend from the clade 21M (according to Nextstrain [38]), called B.1.1.529 within Pangolin [19] and Omicron from the WHO. The set of considered Omicron variants is next referred to

**Table 2. Omicron subvariants described by 1) Pango name (note that all the lineages descending from XBB.1 descendent lineages—including EG.5.1—are recombinants of BA.2.10.1 and BA.2.75 sublineages, i.e., BJ.1 and BM.1.1.1 [41]); 2) Nextstrain clade; 3) Common WHO-given name; 4) Number of characterizing mutations (established by CoVariants.org [33]); 5) Date of first detection, last detection, and number of GISAID sequences (as retrieved from OutBreakInfo.com on 29th of August 2023); 6) Classifications of these variants according to the WHO (retrieved at the end of August 2023)—the ones marked with * were classified as VOCs before 15 March 2023.**

| Variant | Clade | Common name | # Char. Mut. | First detected | Last detected | # Sequences | Classification |
|---------|-------|-------------|--------------|----------------|---------------|-------------|----------------|
| BA.1 | 21K | | 56 | 5-Jan-20 | 27-Jul-23 | 440,101 | * |
| BA.2 | 21L | | 59 | 28-Mar-20 | 10-Aug-23 | 1,240,833 | * |
| BA.4 | 22A | | 66 | 2-Jul-20 | 16-Jun-23 | 38,786 | * |
| BA.5 | 22B | | 61 | 4-Jul-20 | 4-Jul-23 | 25,233 | * |
| BA.2.12.1 | 22C | | 61 | 15-Apr-20 | 29-Jul-23 | 292,096 | - |
| BA.2.75 | 22D | Centaurus | 57 | 8-Aug-20 | 8-Apr-23 | 6,264 | VUM |
| BQ.1 | 22E | Typhon | 68 | 11-Jan-22 | 9-May-23 | 24,572 | - |
| XBB | 22F | Gryphon | 73 | 29-Jun-21 | 15-Aug-23 | 28,607 | VUM |
| XBB.1.5 | 23A | Kraken | 75 | 4-Apr-20 | 15-Aug-23 | 189,754 | VOI |
| XBB.1.16 | 23B | Arcturus | 78 | 23-Feb-22 | 23-Aug-23 | 27,509 | VOI |
| CH.1.1 | 23C | Orthrus | 77 | 9-Jan-22 | 27-Jul-23 | 23,879 | VUM |
| XBB.1.9 | 23D | | 75 | 12-Oct-22 | 20-Jul-23 | 1,581 | VUM |
| XBB.2.3 | 23E | | 76 | 24-Feb-22 | 21-Aug-23 | 5,604 | VUM |
| EG.5.1 | 23F | Eris | 82 | 24-Mar-23 | 23-Aug-23 | 5,727 | VOI |

as *OV*. Then, the list $M_v = m_1, \ldots, m_n$, denoted *list of characterizing mutations* is considered for each variant *v* in *OV*, where $m_i$ is such that at least 75% of viral genomes in *v* exhibit this mutation. We extract $M_v$ for each variant in *OV* as they are reported in OutbreakInfo.org [39]. Fig 2 visually reports the distribution of mutations $M_v$ on Spike for each *v* in *OV* and for the four VOCs observed previously in the pandemic, namely Alpha (B.1.1.7), Beta (B.1.351), Delta (B.1.617.2), and Gamma (P.1).

We note that there are 15 mutations in the Spike protein shared in all the selected variants in *OV*; out of these N764K, D796Y, N856K, Q954H, and N969K are located in the S2 unit of the Spike protein. These five mutations have never been detected in previous VOCs [40]. The difference between the density of mutations in the first rows of the figure is immediately apparent w.r.t. the last four rows (representing previous variants).

## Data integration

Three datasets (two for epitopes and one for variants characterizing mutations) were assessed and integrated, by mapping the characterizing mutations of each variant onto the epitopes

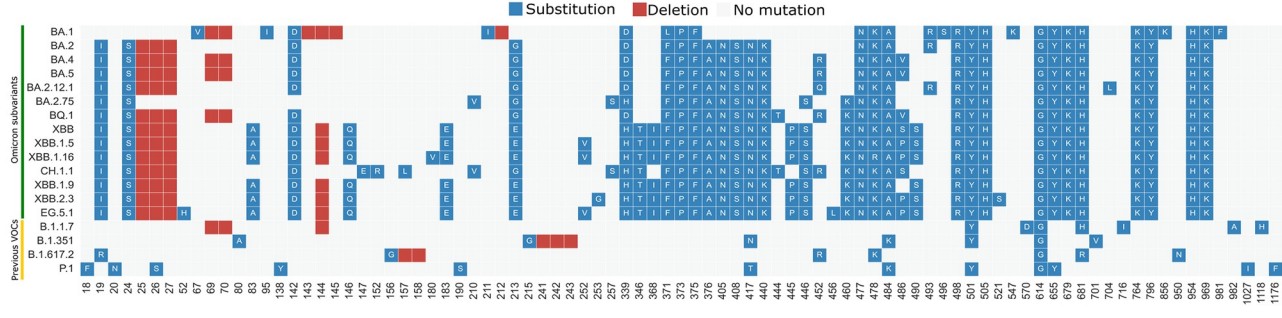

**Fig 2. Overview of Spike protein characterizing mutations for all the variants included in *OV* and previous VOCs Alpha, Beta, Delta, and Gamma.** Blue squares indicate substitutions of the reference amino acid residue with the one indicated by the letter; red squares indicate deletions.

**Table 3. Excerpt of the T cell and B cell linear epitopes coverage table (only Spike protein), including seven centrally-located amino acid positions.** Each cell contains the number of T cell and B cell linear epitopes covering the $p_{n\text{-th}}$ position on the observed protein.

| | $p_1$ | ... | $p_{497}$ | $p_{498}$ | $p_{499}$ | $p_{500}$ | $p_{501}$ | $p_{502}$ | $p_{503}$ | ... | $p_{1273}$ |
|---|---|---|---|---|---|---|---|---|---|---|---|
| T cell epitopes | 7 | ... | 51 | 48 | 46 | 45 | 45 | 43 | 45 | ... | 2 |
| B cell epitopes | 5 | ... | 87 | 83 | 82 | 78 | 78 | 75 | 75 | ... | 9 |

using the corresponding amino acid coordinates. In this process, we used two intermediate data representations.

First, we built a *coverage table* to record the number of T or B cell epitopes that stand on each position of a given protein of SARS-CoV-2. For explanation purposes, Table 3 shows a small excerpt of the Spike protein, where amino acid residues are located in positions $p_1$—$p_{1273}$. As an example, position $p_{498}$ of the Spike protein may exhibit the mutation Q498R; this is a widespread mutation that is considered a *characterizing mutation* of all the 14 variants of *OV*. According to Table 3, this mutation is affecting, respectively, 48 and 83 T cell and B cell linear epitopes. Instead, only 7 T cell (and 5 B cell) epitopes are on $p_1$ and 2 T cell (and 9 B cell) on $p_{1273}$ of the Spike, as a lower density of epitopes was identified on the borders of the protein.

Second, we computed an *array of percentages* of linear T and B cell epitopes affected by each variant in *OV*. Computations are repeated separately for each protein. To achieve so, for a given variant $\bar{v}$ in *OV*, we count the distinct epitopes that are affected by at least one mutation $m$ in the set $M_{\bar{v}}$ of its characterizing mutations; this corresponds to checking that the position of $m$ is included between the start and end position of the epitope. The obtained counts (one for each protein) are finally divided by the total of T cell (respectively, B cell) epitopes. We thus obtained a matrix of percentages for 14 variants by 13 SARS-CoV-2 proteins for T cell epitopes and 12 proteins for B cell epitopes. For instance, T cell epitopes affected by characterizing mutations of BA.1 on the Spike protein are 11.52% of all T cell epitopes on Spike.

## Data aggregation

Each epitope $e$ in IEDB has typically been studied in at least one bibliographic reference, as a result of at least one experimental assay. Thus, many records in the `tcell_full_v3` and `bcell_full_v3` datasets—each representing one assay—refer to the same epitope (i.e., an epitope in the same protein, with the same start-end positions, and therefore the same sequence of amino acid residues), reporting information on different assays that were performed to retrieve the same result.

In our datasets, information that was heterogeneous across the different assays and enclosing references of the same epitope $e$ was aggregated. For a new aggregated record, we considered three columns in the original file, where records represent assays, grouped by reference:

- the "Assay.Qualitative Measurement" (indicating the field Qualitative Measurements in the "Assay" metadata section), representing the qualitative outcome of the assay, which could be described as 'positive', 'positive low', 'positive intermediate', 'positive high', or 'negative',

- the "Assay.Number of Subjects Tested" (called $N$), representing the total number of tested subjects;

- the "Assay.Number of Subjects Positive" (called $R$), representing the total number of subjects that are positive for this assay.

In accordance with the IEDB documentation [42], if an assay lacks information on the number of subjects who responded or tested, we filled in the missing values following the

"Assay.Qualitative Measurement" indication: when an assay outcome is positive, then $N = 1$ and $R = 1$; when an assay outcome is negative, then $N = 1$ and $R = 0$.

For each T cell and B cell linear epitope of the Spike protein, we calculated: the number of positive assays; the number of negative assays; and the response frequency RF (i.e., the number of positively responding subjects divided by the total number of those tested) with its 95% confidence interval (CI). The latter was retrieved according to the following steps:

1. For each reference $r$ regarding $e$, $N_r$ and $R_r$ are the highest values from any existing positive assay of $r$; if there is no positive assay, $N_r$ is the highest N of the negative assays of $r$ and $R_r$ is set to zero.

2. For each epitope $e$, $N_e$ and $R_e$ are the sum of the $N_r$ and $R_r$ of all references regarding $e$.

3. For each epitope $e$, the Response Frequency equals $R_e/N_e$ and its 95% confidence intervals is computed as $95\% CI = \hat{p} \pm 1.96 \sqrt{\hat{p}(1 - \hat{p})/n}$ (as it is distributed as a binomial distribution).

## Data analysis on all epitopes

Two versions of a Monte Carlo-based test were employed to estimate the significance of the placement of characterizing mutations of variants in $OV$ within the set of epitopes. In both cases, for 1,000 repetitions, we randomly repositioned the set of the T cell and B cell linear epitopes extracted from the Spike protein, by sampling their start position on the protein from a uniform distribution. In the first version (called *naive*), for each repetition $i$, we compute $c^i$ as the count of repositioned epitopes affected by at least one mutation of at least one variant in $OV$, thus obtaining the set $C = \{c^1, \ldots, c^{1000}\}$. Then, we calculated the corresponding *p-values* as:

$$p = \frac{|c \in C : c \geq c_o|}{1000}$$

where $c_o$ is the observed number of affected epitopes in the real data.

However, multiple mutations can appear in the same epitope and epitopes vary in length. Thus, we also implemented a second version of the experiment, where these aspects are *weighted*. Considering a set of epitopes $E$, for each $e \in E$ (either real or repositioned), we compute $M_e$ as the count of characterizing mutations of all the variants affecting the epitope—i.e., observed in positions included in the epitope range—and we divide it by the length (number of amino acids) of $e$. Then, for each iteration $i$, we compute the overall factor $f^i$, as the sum of the weighted counts across all the epitopes:

$$f^i = \sum_{e \in E} \frac{M_e}{len(e)}.$$

Again the *p-value* is computed as:

$$p = \frac{|f^i : f^i \geq f_o, i = 1, \ldots 1000|}{1000}$$

being $f_o$ the normalized count computed on the real data.

## Data analysis on highly-responsive epitopes

The same approach was re-applied on a smaller set of T cell and B cell linear Spike epitopes, using an RF threshold of 0.75, which allows us to target only Spike's highly responsive T cell

and B cell linear epitopes. The threshold was selected experimentally; we started with a more strict one and then relaxed it gradually until we got a substantial set of epitopes with high response frequency. S1 Fig presents a histogram of the epitopes RFs.

## Results

As explained in the Methods' Section "Data Integration", we build a coverage table that contains all the proteins included in the epitope datasets. For each amino acid of each protein, we count the number of linear epitopes including that specific position. The coverage of linear epitopes over the Spike protein is shown in the dot plot in Fig 3, where the blue profile reflects B cell epitopes and the orange one reflects T cell epitopes. For each position on the protein (*x*-axis) we show the counts (on *y*-axis) of linear epitopes that overlap that position. As expected, the coverage drops in correspondence to the protein's boundaries, and B cell epitopes have a higher coverage. The Spike protein is cleaved into two subunits, S1 and S2, during the process of viral entry. The S1 subunit contains the N-terminal domain (NTD), which plays a crucial regulatory role in the conformational changes of the Spike [43], and the receptor-binding domain (RBD), which binds to the host cell receptor ACE2, while the S2 subunit facilitates the fusion of the viral and host cell membranes. Various diagnostics, therapeutic agents, and vaccines have been proposed against the Spike protein, with multiple targets, including Spike (RBD, S1-NTD, and less often the S2 subunit) since they are the immunogenic domains of the Spike [44, 45]; however, Omicron subvariants have a high number of characterizing mutations positioned in these regions (especially in the RBD) which makes them more likely to escape immunity.

From Fig 3 it can be noted that both T cell and B cell epitopes are condensed in the RBD region (located in the 319–541 range). In the figure, vertical lines indicate the positions of the characterizing mutations of the selected variants—darker colors indicate that a given mutation is present in multiple variants. A noticeable bundle of characterizing mutations is also affecting the RBD region, hence, affecting the epitopes derived from that region. Also, it can be noted that the NTD region (located in the 13–303 range) has a high number of both T cell and B cell linear epitopes, but it is affected by a wide set of characterizing mutations some of which are deletions (we refer the reader back to Fig 2). Here, two small regions contain peaks of T cell linear epitopes that exceed the counts of B cell linear epitopes; the 142–147 contains many characterizing mutations, whereas the 265–276 range holds none.

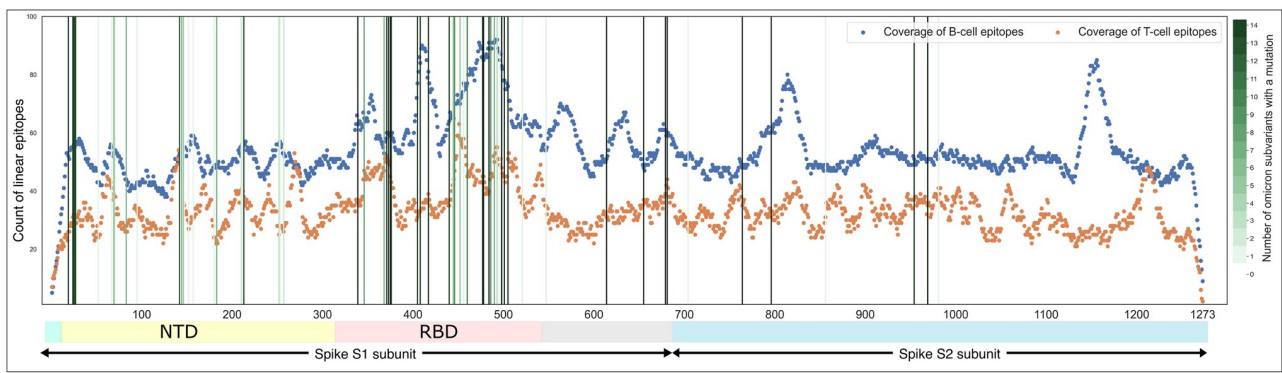

**Fig 3. Coverage of B cell (blue) and T cell (orange) linear epitopes over the Spike protein.** Vertical lines indicate positions of mutations from $M_v$ for all $v \in OV$; the darker the green color, the higher the number of variants with a mutation at that position. The bottom legend indicates the two Spike subunits; we denote NTD: N-terminal Domain (13–303), RBD: Receptor Binding Domain (319–541), and S1/S2 cleavage site (685–686). The full dataset to generate the figure is available as S1 File.

The triple mutations H655Y, N679K, and P681H, found in all Omicron subvariants, are located near the furin cleavage site of the Spike protein and might accelerate the S1-S2 cleavage, translating to more efficient viral entry into the host cell and enhanced infectivity. Another peak of B cell linear epitopes (in the 550–580 range) is located after the RBD region and is not affected by any mutations of Omicron subvariants.

The S2 domain (starting from the 685th amino acid) is quite conserved across all the 14 Omicron subvariants. Two regions in the S2 domain can be noted where many B cell linear epitopes are located (see the two peaks at 800–827 and 1139–1171). Regarding T cell linear epitopes, the counts fluctuate across the S2 domain with a noticeable peak near the C-terminus of the protein (i.e., 1198–1221 range).

Table 4 shows the coverage of T cell and B cell linear epitopes per amino acid per protein. All the structural proteins (S, N, M, and E) have been completely covered with T cell and B cell epitopes, whereas the open reading frames are less dense in this respect. Both B cell and T cell epitopes contribute to the overall immune response and help the immune system effectively combat a wide range of pathogens. However, B cell epitopes are observed to be more abundant than T cell epitopes. This is might be due to the way each type of immune response is functioning (T cell-mediated immunity vs B cell-mediated immunity).

Coverage dot plots of other relevant proteins (N, E, M, and ORF1ab) can be found in S2–S5 Figs, respectively. The Spike has the highest coverage per amino acid compared to all other proteins, which makes it the central target of our study. The other immunogenic protein, the Nucleocapsid, has high coverage per amino acid (covered at least by 4 T cell linear epitopes and 9 B cell linear epitopes). Also M and E proteins, which are notoriously poor in immunogenic activity, are completely covered with both T cell and B cell linear epitopes with considerably high medians compared to other proteins. ORF1ab contains many B cell linear epitopes (median of 29 B cell epitopes covering its amino acids), while it has a very low T cell epitope coverage per amino acid (median of 2 epitopes per amino acid—some regions are not covered with any T cell linear epitopes).

To show the effect of the selected variants over the whole available SARS-CoV-2 set of linear T cell and B cell epitopes, we performed the same analysis on previously circulating VOCs—for these same variants, in Fig 2 we had previously portrayed characterizing mutations of the

**Table 4. Statistics of B and T cell linear epitopes coverage per amino acid (AA), divided by protein.**

| Protein | T cell epitopes per AA | | | | B cell epitopes per AA | | | |
|---------|------|------|---------|--------|------|------|---------|--------|
|         | Min. | Max. | Average | Median | Min. | Max. | Average | Median |
| S       | 2    | 63   | 33.12   | 32     | 5    | 92   | 54.34   | 52     |
| N       | 4    | 34   | 19.53   | 18     | 9    | 56   | 42.21   | 42     |
| M       | 3    | 38   | 20.28   | 20     | 7    | 35   | 28.41   | 29     |
| E       | 3    | 25   | 14.68   | 15     | 5    | 33   | 25.81   | 29     |
| ORF1ab  | 0    | 18   | 2.71    | 2      | 2    | 44   | 29.08   | 29     |
| ORF3a   | 3    | 18   | 8.48    | 8      | 3    | 37   | 25.96   | 25     |
| ORF6    | 0    | 14   | 6.59    | 6      | 2    | 30   | 21.59   | 25     |
| ORF7a   | 1    | 15   | 7.45    | 7      | 3    | 36   | 25.60   | 27     |
| ORF7b   | 0    | 4    | 2.40    | 3      | 2    | 18   | 12.72   | 15     |
| ORF8    | 3    | 12   | 8.25    | 8      | 4    | 43   | 30.40   | 32     |
| ORF9b   | 0    | 5    | 2.59    | 2      | 0    | 1    | 0.43    | 0      |
| ORF9c   | 0    | 5    | 2.14    | 2      | 0    | 0    | 0.00    | 0      |
| ORF10   | 0    | 12   | 4.50    | 2      | 6    | 43   | 25.21   | 27.5   |

**Table 5. Comparative overview of the Spike protein epitopes' coverage of different variants (from previous VOCs and in *OV*).**

| Lineage | # Mutations on Spike | # T cell epitopes | % T cell epitopes | # B cell epitopes | % B cell epitopes |
|---|---|---|---|---|---|
| B.1.1.7 | 10 | 322 | 11.52 | 513 | 11.87 |
| B.1.351 | 10 | 280 | 10.02 | 499 | 11.54 |
| B.1.617.2 | 9 | 256 | 9.16 | 423 | 9.78 |
| P.1 | 12 | 361 | 12.92 | 618 | 14.30 |
| BA.1 | 33 | 731 | 26.16 | 1095 | 25.33 |
| BA.2 | 31 | 695 | 24.87 | 1116 | 25.82 |
| BA.4 | 34 | 776 | 27.77 | 1220 | 28.22 |
| BA.5 | 34 | 776 | 27.77 | 1220 | 28.22 |
| BA.2.12.1 | 33 | 771 | 27.59 | 1216 | 28.13 |
| BA.2.75 | 30 | 722 | 25.84 | 1189 | 27.50 |
| BQ.1 | 36 | 791 | 28.31 | 1253 | 28.98 |
| XBB | 41 | 838 | 29.99 | 1339 | 30.97 |
| XBB.1.5 | 42 | 863 | 30.89 | 1395 | 32.27 |
| XBB.1.16 | 43 | 870 | 31.14 | 1403 | 32.45 |
| CH.1.1 | 41 | 841 | 30.10 | 1340 | 31.00 |
| XBB.1.9 | 40 | 838 | 29.99 | 1339 | 30.97 |
| XBB.2.3 | 43 | 896 | 32.07 | 1454 | 33.63 |
| EG.5.1 | 44 | 894 | 32.00 | 1438 | 33.26 |

Spike protein. Quantitative statistics of the analysis are shown in Table 5. Characterizing mutations of Alpha, Beta, Delta, and Gamma only impacted 9–14% of B/T cell epitopes. Instead, subvariants of Omicron impact 24–34% of B/T cell epitopes. Strikingly, the EG.5.1 (VOI) variant with 44 mutations on Spike impacts 894 (32%) T cell epitopes and 1,438 (33.23%) B cell epitopes, while the XBB.2.3 (VUM) variant with 43 mutations on Spike impacts 896 (32.07%) T cell epitopes and 1,454 (33.63%) B cell epitopes. When compared to the impact of Omicron mutations, percentages of previous Variants of Concern are substantially lower.

As seen in Fig 2, Omicron subvariants shared some characterizing mutations with previous VOCs (e.g., K417N, T478K, and N501Y) even though Omicron is far from these lineages from the phylogenetic point of view. All previously mentioned mutations were related to the ability to destabilize the antibody-binding affinity [46]. Also, Omicron subvariants kept mutating at the same position of previous VOCs but using different amino acid substitutions, e.g., E484K in Beta and Gamma becomes E484A in all Omicron subvariants. Liu et al. [47] found that substitutions at E484 of the Spike were associated with relative resistance to neutralization by several antibodies. However, E484A exhibited higher resistance to more antibodies than other substitutions at the same position (including E484K).

Fig 4 draws the distribution of the linear epitopes of the four structural proteins over the number of variants that exhibit at least one mutation in each epitope. Here, we observe that, out of 3,809 T cell and 5,944 B cell linear epitopes, 1,330 (34.91%) T cell epitopes and 2,178 (36.64%) B cell epitopes have at least one characterizingmutation of any of the selected variants. Instead, 584 (15.33%) T cell epitopes and 941 (15.83%) B cell epitopes are mutated in all 14 selected Omicron subvariants.

On average, T cell linear epitopes on Spike were studied in 1.92 assays, the epitope *YLQPRTFLL* in the range 269–277 being the one studied the most (found in 49 assays, 44 positive and 5 negative assays). On average, B cell linear epitopes on Spike were studied in 1.44 assays. Three epitopes were the most studied ones: *ECDIPIGAGICASYQ* in the 661–675 range studied in 9 positive and 1 negative assays; *NGVEGFNCYFPLQSY* in the 481–495 range

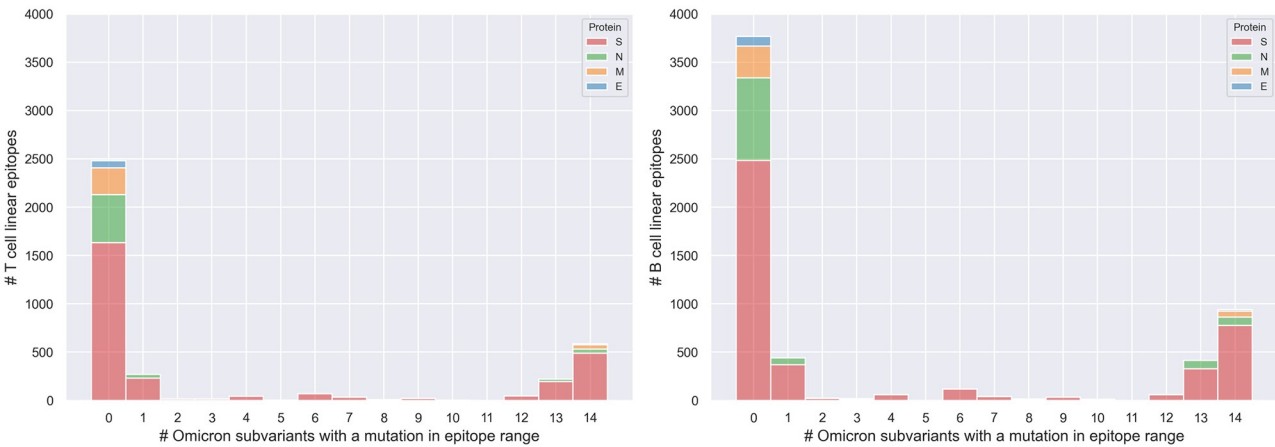

**Fig 4. Distribution of linear T cell (on the left) and B cell (on the right) epitopes, according to the number of variants affecting them.** On the x-axis, we show the number of variants in *OV* whose characterizing mutations (at least one) are exhibited on the observed epitope range. Values regarding different proteins are stacked on the same variant-count bar.

studied in 6 positive and 4 negative assays; and *YNYKLPDDFTGCVIA* in the 421–435 range studied in 5 positive and 5 negative assays.

When aggregating different assays and references' data of different epitopes using their positional amino acid coordinates, we found that T cell linear epitopes of the Spike protein have an average of 27 positive and 37 negative assays per amino acid. Differently, there are 42 positive and 34 negative assays for B cell linear epitopes per amino acid. Fig 5 presents the counts of positive and negative assays for each amino acid of the Spike protein. This visual representation allows us to specify where the regions that have been tested more are located, distinguishing between their positive and negative results. We computed the differences between the positive and negative assay counts per amino acid; in the figure, we show a selection of interesting areas, where a considerable difference between their results is found (i.e., at least 25 assays).

Considering T cell epitopes, a distinctive peak of positive assays is located on 266–279 with the highest value of 146 positive assays specifically located at the 269th amino acid (in line with the peak of the coverage of T cell epitopes mentioned earlier). Also, two regions (593–607 and 1230–1263) were noticed to have positive assays close to zero with a considerably high number of negative assays. In addition, negative assays were more likely to be achieved in the RBD domain.

Considering B cell epitopes, we noticed that the assays' results were comparable over the Spike protein with few exceptions where the count of positive assays overcomes the count of negative assays. A peak of positive assays is located on 785–828 with the highest value of 111 positive assays at the 815-th amino acid and the other two at 555–584 and 1144–1175. These regions correspond to the two peaks found in the coverage plot (see Fig 3), indicating that many B cell epitopes are located in these regions and exhibit positive results when tested; the same was noticed in some ranges of the RBD. The regions 1059–1070 and 1216–1244 have low numbers of positive assays with considerably high numbers of negative assays.

Finally, we calculated the RF and its 95% CI for each T cell and B cell linear epitope. The average RFs per amino acid of the Spike protein are 0.18 (0.13–0.24 95% CI) and 0.21 (0.16–0.26 95% CI) for T cell and B cell epitopes, respectively. The results are shown in Fig 6: here, we observe that the two profiles are relatively distinct; for B cell epitopes five peaks (> 0.40 RF)

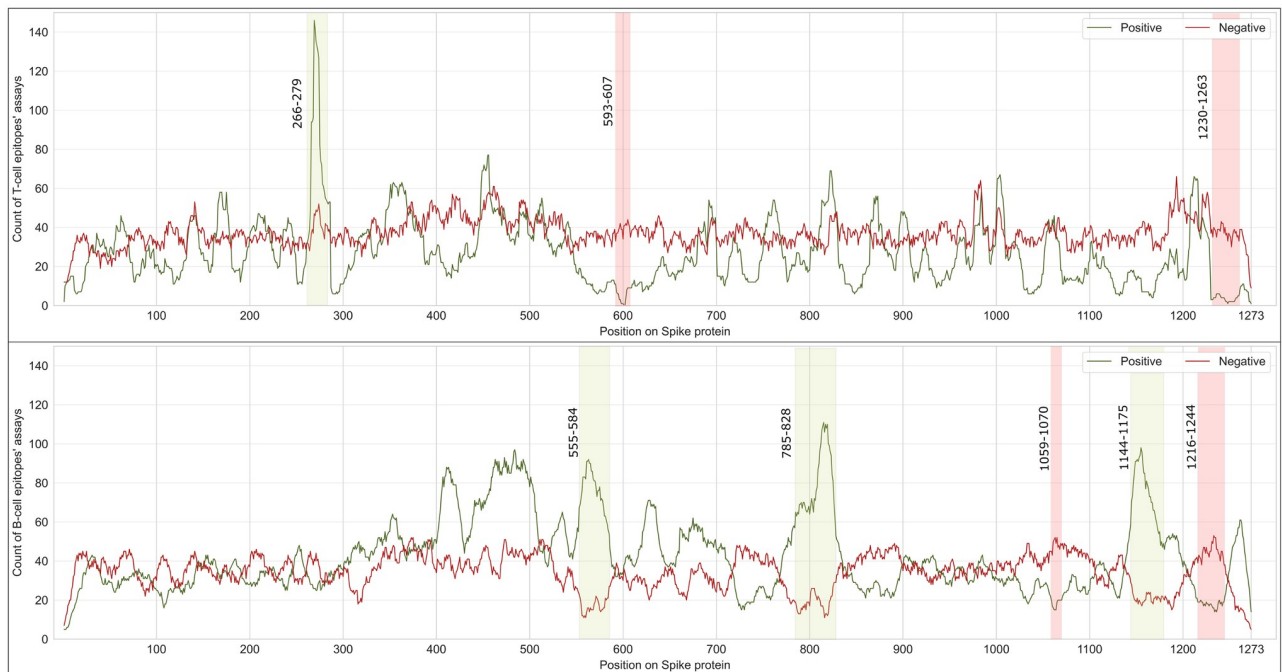

**Fig 5. Counts of positive and negative assays experimental outcome performed on T cell epitopes (top panel) and B cell epitopes (bottom panel) in the Spike protein.** Selected regions with large differences between positive and negative assays are highlighted with green (favoring positive assays) or red (favoring red assays) vertical bands. The full dataset to generate the figure is available as S2 File.

arise around ranges 464–488, 555–584, 793–825, 1145–1162, and 1254–1270. Three of these peaks were also retrieved during the previous steps, meaning that the regions [555–584] in the S1 subunit and [793–825] and [1145–1162] in the S2 subunit of the Spike are rich with B cell linear epitopes (without any Omicron subvariants characterizing mutations) found in many positive assays and linked to high response frequencies. Instead, T cell epitopes have higher RF in the S1 subunit of the Spike protein and lower RF in the S2 subunit. Average RFs are forming a peak hitting 0.47 in the range of 259–280, which does not contain any characterizing mutations of the selected variants. T cell and B cell linear epitopes have diverse RFs across the RBD.

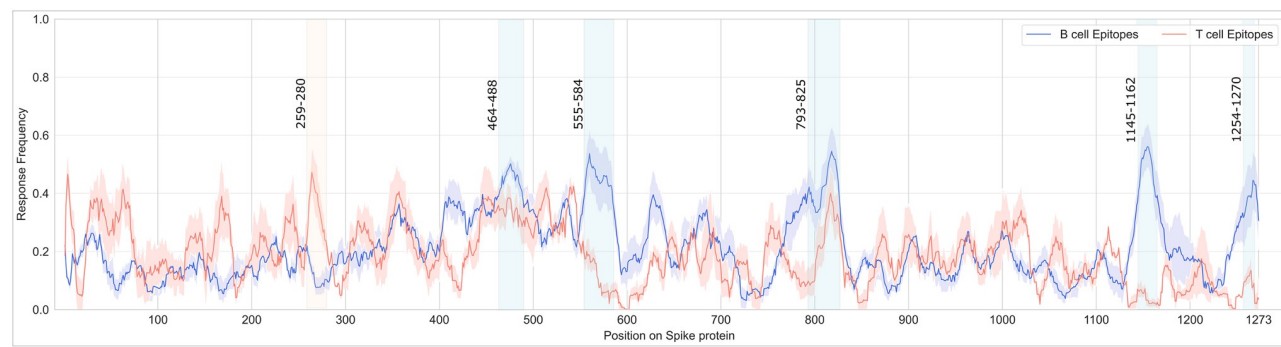

**Fig 6. Average response frequency (RF) flanked by its 95% confidence intervals (CI) of the epitopes computed at each position of the Spike protein.** Regions with high average RF (i.e., peaks > 0.40) are highlighted with red (for T cell) or blue (for B cell) vertical bands. The full dataset to generate the figure is available as S3 File.

Considering B cell epitopes, a wide region with considerably high average RFs (404–505 range) can be noticed; for T cell epitopes, two peaks in the RBD domain are detected (348–364 and 443–486 ranges).

Note that, the integration of the intervals detected in Figs 5 and 6, allows us to identify regions in the Spike protein with both a high number of positive assays and high response frequencies; such a preliminary filter could be a recommended first step when selecting the target region of antibody during its design/development.

## Monte Carlo simulation results

Next, we employed the Monte Carlo simulation approach with the objective of determining the significance of identifying 1,163 T cell epitopes and 1,840 B cell epitopes affected by at least one of the characterizing mutations, among a total of 2,794 and 4,323 epitopes, respectively.

For 1,000 times we randomly re-positioned all the Spike linear epitopes by sampling their start position on the Spike protein from a uniform distribution. In the *naive* version, for each configuration, we compute the count of epitopes that are affected by at least one characterizing mutation of at least one Omicron subvariant and retrieve the distribution of such counts. Fig 7 presents two histograms illustrating the distributions of the counts of impacted T cell and B cell epitopes across the 1,000 simulations. We compared those distributions with the *observed* count (see red line). We compute the *p-values* as the number of random conformation having a count greater or equal than the observed one, obtaining *p-values* <0.001 for both T cell and B cell linear epitopes.

In the *weighted* version, we weigh the count of the epitopes by the number of mutations affecting each epitope, normalized by the epitope length. Similarly to the first case, also here *p-values* are < 0.001 for T cell and B cell Spike linear epitopes. The related plot is provided in the S6 Fig.

## Highly-responsive epitopes results

Finally, we focused on a dataset of highly responsive epitopes, derived from the original Spike linear epitopes dataset, using RF > 0.75 as a threshold for responsiveness; here, we retained 370 and 706 T cell and B cell epitopes, respectively. For this specific set, we first show the

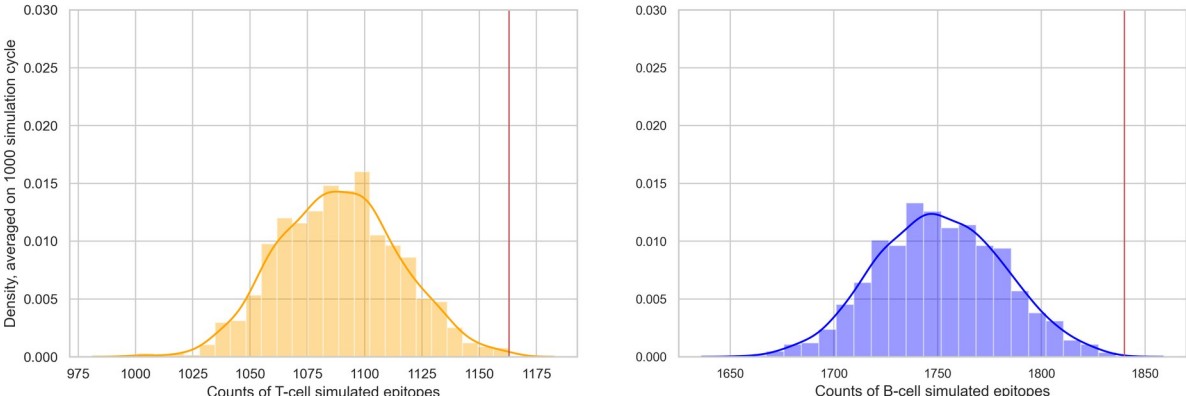

**Fig 7. Monte Carlo-based test results, considering the *naive* test version.** We plot the density distribution of T-cell (left) or B-cell (right) simulated epitopes affected by at least one sampled mutation (randomly selected from all Omicron subvariants characterizing mutations). The red vertical lines (at 1,163 T cell and 1,840 B cell epitopes) represent the observed count of linear epitopes from the Spike protein that have been mutated.

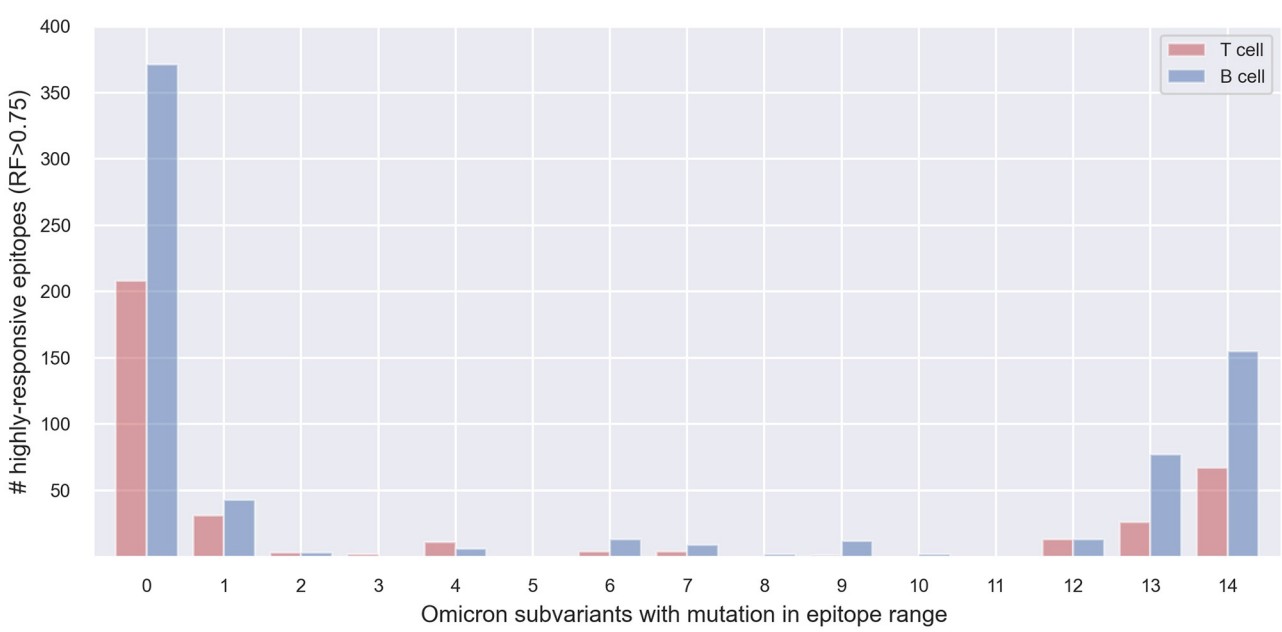

**Fig 8. Distribution of highly-responsive epitopes (i.e., with RF >0.75) according to the number of variants that affect them.**

distribution of the selected epitopes over the number of variants that have at least one mutation in each epitope (see Fig 8). Their positive and negative assay counts and average RFs per amino acid of the Spike protein are provided in S7 and S8 Figs. We observe that 162 (43.78%) selected T cell epitopes and 335 (47.45%) selected B cell epitopes have at least one characterizing mutation of any of the selected variants, meaning that they appear in the 1–14 positions in the barplot. Instead, 67 (18.10%) selected T cell epitopes and 155 (21.95%) selected B cell epitopes are mutated in all 14 Omicron subvariants (*OV*).

## Monte Carlo simulation results

When running the Monte Carlo-based test, on data with simulated positions, using the *naive* version (plain counts of mutated re-positioned epitopes), we obtained *p-values* equal to 0.037 and $< 0.001$ for the subsets of T cell and B cell epitopes, respectively. The related plot is provided in the S9 Fig. In the *weighted* version (weighted counts of mutated re-positioned epitopes), we obtained *p-values* equal to 0.004 and $< 0.001$ for the subsets of T cell and B cell epitopes, respectively (see Fig 9). A slight increase in the (still significant) *p-values* was observed when the test was performed on smaller T cell epitopes subsets, possibly due to the considerable decrease in the number of tested epitopes.

## Analysis reproducibility: Focusing on the most recent variants

The proposed analysis workflow is completely reproducible. The code (fully disclosed and documented on our Zenodo public repository [48]) can be rerun at any point in time in the future. To demonstrate this, we reran the pipeline by considering SARS-CoV-2 variants that have risen at a later time than the original analysis run in August 2023. Assuming that the IEDB resource will be maintained for a long time, by downloading updated input datasets, any reader is able to regenerate all the numerical and visual artifacts of the previous analysis.

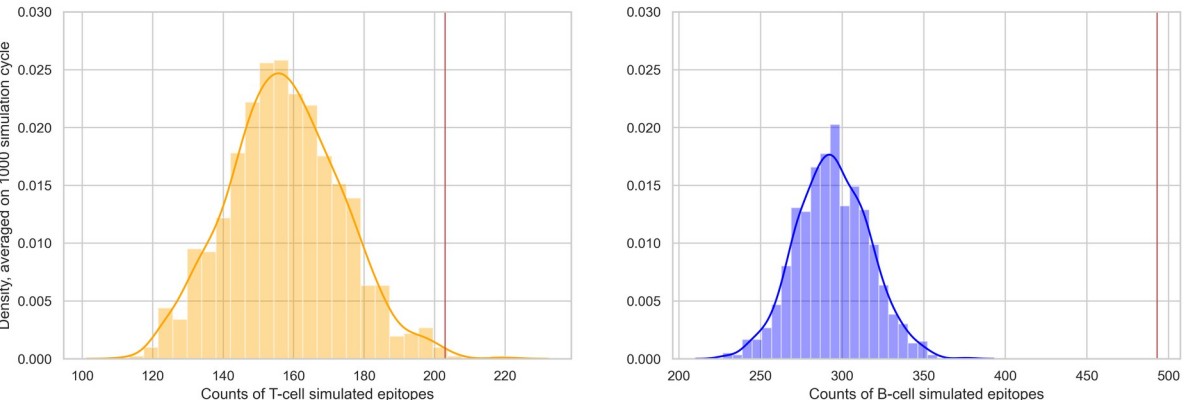

**Fig 9. Monte Carlo-based test results, simulating epitopes with RF > 0.75 and employing the weighted test version.** We plot the density distribution of T-cell (left) or B-cell (right) simulated epitopes' factors (see Methods for details), affected by at least one sampled mutation. The red vertical lines (at 203.15 T cell and 493.75 B cell epitopes) represent the observed sum of computed factors for all the epitopes in the selected subset.

Here, we describe a selected portion of the most interesting results. Four additional Omicron subvariants (XBB.1.5.70, HK.3, BA.2.86, and JN.1) were tested over the IEDB T cell and B cell datasets (retrieved on 28, May 2024). It can be observed that the JN.1 and its ancestor BA.2.86 (with their 60 and 58 characterizing mutations in the Spike protein, respectively), both affect 41,29% and 43,07% of the Spike T-cell and B-cell linear epitopes, respectively. Instead, HK.3 (with its 45 characterizing mutations) affects 32,81% and 33,59% of the Spike T-cell and B-cell linear epitopes, respectively. Lastly, XBB.1.5.70 has 44 characterizing mutations on the Spike protein and affects 32,00% and 32,66% of the Spike T-cell and B-cell linear epitopes, respectively.

It is worth mentioning that both JN.1 and BA.2.86 are classified as VOIs by WHO (as of 3 May 2024); also, both share 15 distinct Spike characterizing mutations (e.g., S50L in NTD and N450D in the RBD which are thought to mediates resistance to some monoclonal antibodies [49]), which were not detected in any other omicron subvariants listed in our analysis. But, with its additional Spike mutation L455S, JN.1 rapidly achieved extensive resistance across antibodies targeting the RBD of the Spike and showed higher immune evasion compared to BA.2.86 [50]. On the other hand, both variants XBB.1.5.70 and HK.3 have L455F and F456L characterizing mutations on their RBD of the Spike protein, which is known as the '*FLip*' genotype, so-called because of the abbreviations of the amino acids involved and the fact that the location of the *F* and *L* amino acids is switched; the '*FLip*' genotype is associated with immune escape properties too [51–53].

## Discussion

*Comparison with related work.* Several tools for epitope prediction have been studied in the past [54]; a series of projects were dedicated specifically to SARS-CoV-2 epitopes. COVIEdb [55] targets pan-coronavirus vaccine development; here, SARS-CoV-2 database entries were predicted by using tools hosted by IEDB, exploiting the similarity of other viruses. DBCOVP [56] and the COVID profiler [57] provide companion vaccine design tools, with a focus on prediction, with light integration with IEDB data. A small number of tools are specifically dedicated to the analysis of epitopes conservancy. The IEDB Epitope Conservancy Analysis (ECA) tool [58] requires inputs from users for both epitopes and mutations. COVIDep [59] is an

integrative effort joining IEDB epitopes with GISAID sequences, with a "Population coverage analysis" providing quantifications of "conservation" and "population coverage" for each epitope. Our tool EpiSurf [31] also allows for conservancy and population coverage analysis, with the possibility to select a user-defined sequence population. Unfortunately, both COVIDep and EpiSurf have discontinued the updates of new sequenced in September 2021 due to the huge growth of the GISAID database. The Virus Pathogen Database and Analysis Resource (ViPR) [60] connects both predicted and experimentally derived epitopes and proteins of sequences deposited on GenBank. Finally, the COG-UK Mutation Explorer (COG-UK-ME) [9] presents a UK-centred interface to UK genomes and variation (also in the context of T cell epitopes reported by experimental studies). COVIDep, ViPR, and COG-UK-ME currently offer a curated list of epitopes, respectively predicted from SARS-CoV, predicted with NetCTL [61], and manually extracted from experimental studies.

Regrettably, none of these tools were fit for the purpose targeted in this analysis, which had to be completely re-engineered with an ad hoc data science pipeline. Here, we proposed to study the entire database of SARS-CoV-2 epitopes curated by IEDB to understand how epitopes are distributed. Specifically, we did not make any assumptions on the possible effects of the studied viral mutations on the immune response itself, but we separated the analysis for each type of immune cell; T cell (including both CD4+ and CD8+ cells) corresponds to cellular immunity and B cell corresponds to humoral immunity.

*Contributions.*

Overall, our key contributions are:

1. *Epitopes' distribution analysis.* We provide insights into the coverage of SARS-CoV-2 viral proteins by T cell and B cell epitopes (Table 4), highlighting the regions with high-density mutations (Fig 3 and S2–S5 Figs), and aiding in identifying conserved epitopes crucial for vaccine design.

2. *Impact of omicron characterizing mutations.* Our study details how mutations in various Omicron (and non-Omicron) variants impact T cell and B cell epitopes (Table 5 and Fig 4). This information is essential for understanding the evolving landscape of SARS-CoV-2 and its implications for immune recognition.

3. *Key Spike protein region identification.* By analyzing the response frequency of assays related to specific epitopes, we identify key regions in the Spike protein (Figs 5 and 6). These include the S2 domain and specific segments of the S1 subunit, that remain conserved across multiple Omicron subvariants. These regions could be promising targets for therapeutic antibodies and vaccine development.

We finally confirmed our results by performing a Monte Carlo simulation on the entire dataset (Fig 7) and on high RF epitopes (Figs 8 and 9). Our pipeline is completely reproducible, as shown in a dedicated subsection, updating our results to May 2024.

## Limitations

Our study also presents a number of limitations. Regarding the employed datasets: even though IEDB is currently the largest open-access database for reporting epitopes, their datasets only represent the epitopes studied in the literature. This may result in a bias towards regions of the viral proteins that have been more intensively studied, potentially overlooking epitopes in less examined areas. Also, the variability in sample sizes across different studies could affect the robustness of our results. Moreover, the immune response data is subject to assay biases. Variations in assay sensitivity, specificity, and the conditions under which they were

performed can influence the detection of epitopes and response frequencies. This may impact the generalizability of our findings to broader populations and different experimental settings. In addition, we chose to include epitopes from any species, as our aim is to indicate how the already experimentally studied epitopes are affected by the currently circulating variants regardless of their host or the immune response pathway that these epitopes are involved in.

## Significance

When designing vaccines, researchers aim to target epitopes that are highly immunogenic and conserved across different pathogen strains. This helps ensure the vaccine will be effective in a broad population and provide long-lasting protection against the target pathogen. High response frequency to a specific epitope suggests that the epitope is a good candidate for inclusion in a vaccine, as it is more likely to generate a strong and effective immune response in a larger portion of the population. This can contribute to developing more effective vaccines against various infectious diseases. Since Omicron has been heavily studied for its ability to permit evasion of vaccine-induced immunity [24, 62, 63]; in our study, we aimed to identify the regions where a high number of positively detected highly responsive epitopes are found, which might be good vaccine/therapeutic target candidates. For example, we observed that the S2 domain is more conserved across all the 14 Omicron subvariants, suggesting it could be a good target for therapeutic antibodies and vaccines, which is in line with what Guo et al. have suggested [64]. Three distinctive regions (555–584, 800–825, and 1145–1162) of the S2 subunit with high B cell epitopes, linked to many positive assays and high RF were noticed without any characterizing mutations of any of the selected Omicron subvariants (see Fig 6).

Interestingly, within the NTD region of the S1 subunit (specifically, 259–280 range) a high number of T cell epitopes were found and studied in a high number of positive assays (resulting in high RFs); this small region does not have any characterizing mutations of the Omicron subvariants.

Omicron is continuously evolving under immune pressure [65], and selection pressure may drive viral mutations for enhanced immune evasion [66]. Compared to previous VOCs, selective immune pressure was noticed by Duerr et al. [63] with the additional booster vaccine shots against the Delta variant during the period of an evolutionary transition from Delta to Omicron BA.1-BA.5. But, some of the variants included in the considered *OV* set (BA.2 and BA.5) were found to have a selective advantage under booster vaccination pressure, contributing to the evolution of BA.2 and BA.5 subvariants and recombinant forms that predominate in 2023 in which multiple sites underwent strong positive selection with a particular focus on the receptor-binding motif (RBM), including sites 417, 440, 444, 452, 486, and 493 that represent the most immune evasive mutation sites according to Duerr et al. [63], most of those mutations are found in almost all of the variants in our *OV* (see Fig 2). Other cases of selective immune pressure were also found in other omicron variants included in *OV* [67–69].

## Conclusions

We conducted a database-wide analysis on the impact of lineages' characterizing mutations on all T cell and B cell linear epitopes collected in the Immune Epitope Database (IEDB) for SARS-CoV-2 focusing on the omicron subvariants. We presented a workflow of this analysis and confirmed our results by performing a Monte Carlo simulation on both the entire dataset and selected epitopes with high response frequencies. Our pipeline delivers a comprehensive analysis of how mutations in omicron subvariants impact T cell and B cell epitopes. This approach focuses on epitopes distribution and mutation impact without pre-assumptions on

immune response effects and extends to study the response frequency and assays of these epitopes.

The identification of highly immunogenic and conserved epitopes can inform the design of vaccines that offer broad and long-lasting protection. Our findings suggest specific regions of the Spike protein that are optimal for inclusion in vaccine formulations.

We note that three of the variants with the highest percentages of affected T cell and B cell Spike epitopes are XBB.1.5, XBB.1.16, and EG.5.1 (see Table 5). These variants are the only ones harboring G252V mutation (see Fig 2), which is linked to increased neutralization resistance of these variants [70]. Also, F486 on the Spike protein seems to be a hotspot for mutations with different AA alternatives depending on the variants (F486V in BA.4, BA.5, and BQ.1; F486S in XBB and CH.1.1; and F486P in XBB.1.5, XBB.1.16, XBB.2.3, and EG.5.1). Our analysis allowed us to pinpoint these locations and suggest that these locations are carefully studied before epitopes including them are designed.

The WHO has revised the nomenclature in October 2023 [13]. All variants included in *OV* are currently considered VOIs (XBB.1.5 and XBB.1.16) or VUMs, or de-escalated variants meaning that, at some point, they were considered as VOCs or VOIs by WHO, CDC, and/or ECDC. According to ECDC, a variant could be de-escalated based on at least one of the following criteria: (1) the variant is no longer circulating, (2) the variant has been circulating for a long time without any impact on the overall epidemiological situation, (3) scientific evidence demonstrates that the variant is not associated with any concerning properties. Designations may continue changing, contributing to making this effort even more relevant as continued monitoring becomes necessary to understand the implications of variants on epitopes. Then, SARS-CoV-2 warning systems could include not only the ability to recognize novel variants at their early stages [28, 29] but also a report on which epitopes are impacted by the arising mutations and suggested implications, in the form that we proposed in this study. Automatization of these analyses is expected in our upcoming work.

## Supporting information

**S1 Fig. Histograms showing the counts of epitopes (y-axis) with calculated RFs (x-axis), for T cell (left) and B cell (right) Spike linear epitopes.** The vertical red line (RF = 0.75) is the selected threshold for filtering both cases.
(TIF)

**S2 Fig. Coverage of B cell (blue) and T cell (orange) linear epitopes over the Nucleocapsid protein.** Vertical lines indicate positions of mutations from $M_v$ for all $v \in OV$; the darker the green color, the higher the number of variants with a mutation at that position.
(TIF)

**S3 Fig. Coverage of B cell (blue) and T cell (orange) linear epitopes over the Envelope protein.** Vertical lines indicate positions of mutations from $M_v$ for all $v \in OV$; the darker the green color, the higher the number of variants with a mutation at that position.
(TIF)

**S4 Fig. Coverage of B cell (blue) and T cell (orange) linear epitopes over the Membrane protein.** Vertical lines indicate positions of mutations from $M_v$ for all $v \in OV$; the darker the green color, the higher the number of variants with a mutation at that position.
(TIF)

**S5 Fig. Coverage of B cell (blue) and T cell (orange) linear epitopes over the ORF1ab polyprotein.** Vertical lines indicate positions of mutations from $M_v$ for all $v \in OV$; the darker the

green color, the higher the number of variants with a mutation at that position.
(TIF)

**S6 Fig. Monte Carlo-based test results, considering the weighted test version.** We plot the density distribution of T-cell (left) or B-cell (right) simulated epitopes' factors (see Methods for details), affected by at least one sampled mutation. The red vertical lines (at 1353 T cell and 2071 B cell epitopes) represent the observed sum of the computed factors of all linear epitopes from the Spike protein that have been mutated.
(TIF)

**S7 Fig. Counts of positive and negative assays performed on T cell (top) and B cell (bottom) Spike highly responsive epitopes, derived from the original Spike linear epitopes dataset, using RF > 0.75 as a threshold for responsiveness.**
(TIF)

**S8 Fig. Average response frequency (RF) flanked by its 95% confidence intervals (CI) of the highly responsive epitopes computed at each position of the Spike protein.**
(TIF)

**S9 Fig. Monte Carlo-based test results, simulating epitopes with RF > 0.75 and considering the *naive* test version.** We plot the density distribution of T-cell (left) or B-cell (right) simulated epitopes affected by at least one sampled mutation. The red vertical lines (at 162 T cell and 335 B cell epitopes) represent the observed count of linear epitopes from the epitopes in the selected subset of Spike protein that have been mutated.
(TIF)

**S1 File. Dataset used to generate Fig 3.** For each amino acid of the spike protein, it reports T cell and B cell linear epitope counts and the number of omicron subvariants from the *OV* set with at least a characterizing mutation affecting that amino acid.
(XLSX)

**S2 File. Dataset used to generate Fig 5.** For each amino acid of the spike protein, it reports the positive and negative assay experimental counts performed on T cell and B cell spike linear epitopes and calculated as the sum of overlapping linear epitopes with the observed position.
(XLSX)

**S3 File. Dataset used to generate Fig 6.** For each amino acid of the spike protein, it reports the average RF values and 95% CI (lower and upper bound) of the linear T cell and B cell epitopes.
(XLSX)

## Acknowledgments

The authors would like to thank Prof. Stefano Ceri and Prof. Matteo Chiara for the fruitful discussions that inspired this work.

## Author Contributions

**Conceptualization:** Anna Bernasconi, Pietro Pinoli.

**Data curation:** Ruba Al Khalaf.

**Formal analysis:** Ruba Al Khalaf.

**Funding acquisition:** Anna Bernasconi.

**Investigation:** Ruba Al Khalaf.

**Methodology:** Anna Bernasconi, Pietro Pinoli.

**Project administration:** Anna Bernasconi.

**Software:** Ruba Al Khalaf.

**Supervision:** Anna Bernasconi, Pietro Pinoli.

**Writing – original draft:** Ruba Al Khalaf, Anna Bernasconi.

**Writing – review & editing:** Anna Bernasconi, Pietro Pinoli.

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
