## [Decision Letter · Decision Letter 0]

15 May 2024

PONE-D-24-11657Systematic analysis of SARS-CoV-2 Omicron subvariants’ impact on B and T cell epitopesPLOS ONE

Dear Dr. Bernasconi,

Thank you for submitting your manuscript to PLOS ONE. After careful consideration, we feel that it has merit but does not fully meet PLOS ONE’s publication criteria as it currently stands. Therefore, we invite you to submit a revised version of the manuscript that addresses the points raised during the review process.

We look forward to receiving your revised manuscript.

Kind regards,

Nagarajan Raju

Academic Editor

PLOS ONE

"This work has been funded by Ministero dell’Università della Ricerca (PRIN PNRR 2022 “SENSIBLE” project, n. P2022CNN2J), Principal Investigator: Anna Bernasconi."

4. We notice that your supplementary figures are included in the manuscript file. Please remove them and upload them with the file type 'Supporting Information'. Please ensure that each Supporting Information file has a legend listed in the manuscript after the references list.

Additional Editor Comments:

I suggest authors to go through the comments and address them in the revised version of the manuscript

Reviewers' comments:

Reviewer's Responses to Questions

**Comments to the Author**

1. Is the manuscript technically sound, and do the data support the conclusions?

Reviewer #1: Yes

Reviewer #2: Yes

Reviewer #3: Yes

2. Has the statistical analysis been performed appropriately and rigorously? 

Reviewer #1: Yes

Reviewer #2: Yes

Reviewer #3: Yes

3. Have the authors made all data underlying the findings in their manuscript fully available?

Reviewer #1: Yes

Reviewer #2: Yes

Reviewer #3: Yes

4. Is the manuscript presented in an intelligible fashion and written in standard English?

Reviewer #1: Yes

Reviewer #2: Yes

Reviewer #3: Yes

5. Review Comments to the Author

Reviewer #1: The scientific rationale and investigations are sound. Monte Carlo analysis was used as a statistical test of significance. It appears that a complete upload of underlying data and workflow is present, but I suggest upload of the data from one figure to aid potential analysis by future readers (see review comment 4). While the data is presented and described free of spelling and grammatical errors, improvement in the clarity and presentation of the article is needed before acceptance (review comments 1-3, 5).

Reviewer #2: The study entitled “Systematic analysis of SARS-CoV-2 Omicron subvariants’ impact on B and T cell epitopes” presents an extensive analysis of the potential impact of mutations within Omicron subvariants on B- and T-cell epitopes, which is crucial for vaccine and monoclonal antibody development. While this paper makes a significant contribution to the field, it is essential that the information and evidence are well-structured and articulated. Therefore, I believe it is suitable for publication in PLOS ONE.

Some considerations regarding the following minor points:

- Change the short title, as it is identical to the main title.

- The abstract provides relevant introductory information; however, no results are mentioned. Please include the main findings of the paper and any future direction.

- Line 21: “as a candidate to become the most dangerous variant of SARS-CoV-2” appears to be a vague statement. Please rephrase for clarity or consider removing it.

- Check acronyms for consistency. When using an acronym, introduce the full term on first mention, followed by the acronym in brackets, to be used thereafter.

- I am concerned about the timing of retrieving epitope datasets and variant information, which dates back to August 2023, resulting in a lag of over 8 months so far. Could you provide any updates, if possible, at least on the current variants (e.g., JN.1 and its subvariants)?

- Considering the length of the materials and methods section, please include only the relevant information in the main text. Add to supplementary.

- In the discussion, provide additional context and background on the role of immunological pressure and proofreading activity on the epitopes impacted by Omicron mutations.

- What is the implication of the study's finding on antibody recognition and monoclonal antibody development?

- What is the added value of this paper in the current context?

- As the conclusion is integrated within the discussion, please briefly highlight your perspective and the potential value of the paper's findings.

Reviewer #3: This study presents a comprehensive analysis of how mutations in the Omicron variants of SARS-CoV-2 impact T cell and B cell linear epitopes. Utilizing data from the Immune Epitope Database (IEDB), and tried to develop robust analysis pipeline to investigate the distribution of epitopes affected by Omicron-characterizing mutations. The findings highlight novel insights into the coverage of viral proteins by epitopes, the distribution of mutated epitopes across Omicron variants, and the impact of Omicron mutations on epitope recognition. This study somehow provid a detailed understanding of the interplay between viral mutations and immune epitopes, contributes valuable information for future genomic surveillance and epidemic response strategies. The manuscript is acceptable in current form howler it might be advantage to state and discuss following issues with extra detail

1- It would be helpful to provide more context on the significance of the findings in relation to vaccine development and immune response.

2- Consider discussing potential limitations of the study, such as sample size variations or assay biases, and how they may have influenced the results.

6. PLOS authors have the option to publish the peer review history of their article (what does this mean?). If published, this will include your full peer review and any attached files.

Reviewer #1: No

Reviewer #2: **Yes: **Mohammad Alkhatib

Reviewer #3: No

---

## [Author Response · Author response to Decision Letter 0]

26 Jun 2024

Please see the "response to reviewers.pdf" file. 

Dear Dr. Nagarajan Raju, 

We thank you for coordinating the review process for our manuscript, and the three reviewers for their time and efforts in reading and evaluating our work while providing their comments and suggestions. We are glad that the presented analysis was, in general, positively received.

Please find attached the revised manuscript (where changes are marked in red color). Note that our key revisions include the following.

As required by Reviewer #1:

1. Complete revision of all Tables and Figures’ captions to make them more self-explanatory;

2. Change in the structure of data analysis section, where first the complete epitopes dataset is employed, then we restrict our analysis to only highly-responsive epitopes;

3. Explicit mention of what parts of the manuscript use real data and what parts employ simulated data;

4. Provision of three new Supplementary Files for regenerating Figures 3, 5, and 6.

As required by Reviewer #2:

5. More explicit contribution in the abstract;

6. New section on reproducibility and results on recent variants;

7. Reorganization of the Discussion section, highlighting our proposed work’ impact and value.

As required by Reviewer #3:

8. In the discussion, brief hint on implications on vaccine development and immune response;

9. In the discussion, new paragraph on our approach limitations.

Below, we provide a detailed point-by-point response to each reviewer's comment. 

We believe that our paper, after careful and complete revision, has greatly improved and deserves publication in PLOS ONE. Thank you for the opportunity to revise and resubmit our manuscript. We appreciate the valuable feedback provided by the reviewers and the editorial team. We look forward to hearing from you regarding the status of our submission. 

With best regards, 

Anna Bernasconi 

(corresponding author)

Reviewer #1

>>>Reviewer #1 (text from doc)

The submitted research article by Khalaf et al. explores the localization of SARS-CoV-2 Omicron variant amino acid mutations within T cell and B cell epitopes collected in the Immune Epitope Database. The work leverages a data-science approach and provides a nice demonstration of analyzing and curating publicly available data. The analyses are appropriate and performed well, but some of their presentation led to confusion. I mark all these issues as ‘minor,’ because I don’t believe any new experiments or re-analysis will need to be performed. Nevertheless, I believe that modification/clarification of the text is critical for publication and to allow the hard work of this group to be clearly presented.

>>>Our response

We thank the Reviewer for positively evaluating our analyses and providing suggestions for better presenting our work. In the revised manuscript, we tried our best efforts to remove any source of confusion and clarify all the relevant steps.

>>>Reviewer #1 (text from doc)

1. Figure legends need to be addressed. Figure legends should enable the figure to allow the reader to reasonably understand the data independent of the main text. Figure 9 is the biggest example. I will not clarify every point, but most figures (and supplementary figures) would benefit from improved description. A couple of specific points:

a. Figure 8, also clarifying this was from simulated data (is it? See below), to prevent confusion with data from Figure 4.

b. Figure 5 (and methodology/main text [Line 340]): what defines “considerable differences” that warrants highlighting

>>>Our response 

Thank you for highlighting this issue. In the revised version of the manuscript, we extended most of the figures’ captions (specifically, Figures 3-9, Tables 1, 4-5, and all Supplementary figures), using a more comprehensive style; we believe that now figures’ captions are more self-contained and understandable by readers.

Considering point (a), the data in Figure 8 is real data (not simulated): it was generated using a filter on the calculated response frequency (as it is now highlighted in the caption). Thanks to the Reviewer’s comment, we realized that the organization of the Results section could lead to some confusion. We now first report all the Results related to the full original epitopes dataset (up to Figure 6), validated with a Monte Carlo simulation (i.e., Fig. 7), up to Page 14, line 398. 

After, we separately discuss the same analysis restricted to a smaller dataset, only including highly-responsive epitopes (see new subsection ’Highly-responsive epitopes results’ marked on Page 14, lines 399), where we first provide a description of the data distribution (Figure 8), then, we run a Monte Carlo test on simulated data (shown in Figure 9) -- this paragraph concludes on Page 15, line 420.

Considering point (b), we calculated the difference between the assays counts per Spike amino acids and detected the peaks in the difference (i.e., where positive and negative counts differ by at least 25 assays in absolute number). We reported in the manuscript only a relevant selection of these areas. The full dataset to generate the figure is now made available as Supplementary File S2. In the revised version of the manuscript, we rephrased this aspect for better clarity (see Page 12, lines 343-345). Similarly, we now make available Supplementary File S3, used to generate Figure 6.

>>>Reviewer #1 (text from doc) 

2. Calling the two different Monte Carlo methodologies “variants” is confusing, since variant is used so frequently used to refer to Omicron lineages. Would suggest finding another phrase to describe the two simulations.

>>>Our response

We understand the concern of the reviewer and apologize for the confusion. We consequently opted for using a different term (i.e., “version”) when referring to the two Monte Carlo methodologies. 

>>>Reviewer #1 (text from doc) 

3. Lines 391-400. I am confused by this section, and Figure 8. I can’t tell if this is ‘real’ data that has been included in the Monte Carlo simulation section, or simulated data that is not clearly described, or something else that I am misunderstanding. Is this data effectively the data from Figure 4, but using only RF>0.75 epitopes? I think the transition from simulated data to real data within the same section is creating some confusion.

>>>Our response 

We apologize for the confusion in this section. Please note that Figure 8 shows the distribution of the real Spike linear highly responsive epitopes (with RF > 0.75) according to the number of selected Omicron subvariants with at least one characterizing mutation affecting the epitope range. As mentioned in the response to point (1a) of this Reviewer, we re-organized the Results for avoiding this confusion..

More specifically, first, in the Results we included all the plots and descriptions related to the full original epitopes dataset (validated with a Monte Carlo simulation, i.e., Fig. 7). 

Separately, we discuss the restriction to a smaller dataset, only including highly-responsive epitopes (see new subsection, on Pages 14-15, lines 399-410). Here, we first provide a description of the data distribution (Figure 8), then we briefly comment on the following results (which, for space concerns, have been moved to the Supplementary Figures 7, 8 and 9). Also in this case, we produce a Monte Carlo test on simulated data (shown in Figure 9). 

This structure has been explained and reinforced in the first paragraph of the Materials and Methods section (see Page 3, lines 81-82). We trust that, with the new organization, the steps will be clearer.

>>>Reviewer #1 (text from doc) 

4. If possible, including XY coordinate values of Figure 3 as an Excel/CSV file in the supplemental data may be helpful. This could allow researchers to quickly manipulate the data and find regions with high epitope coverage more easily.

>>>Our response 

We appreciate the suggestion of the reviewer and now make available coordinate values to generate Figure 3 as a CSV file (Supplementary File S1), to allow readers to investigate further details on Spike linear epitopes coverage. Following the same reasons, we now make available Supplementary Files S2 and S3, used to generate Figures 5 and 6, respectively.

>>>Reviewer #1 (text from doc) 

5.Line 55: Change “contrast” to “combat”

>>>Our response

Thank you for suggesting this replacement. We duly applied it in the revised version of the manuscript. 

>>> Reviewer #1 (text from form)

The scientific rationale and investigations are sound. Monte Carlo analysis was used as a statistical test of significance. It appears that a complete upload of underlying data and workflow is present, but I suggest upload of the data from one figure to aid potential analysis by future readers (see review comment 4). While the data is presented and described free of spelling and grammatical errors, improvement in the clarity and presentation of the article is needed before acceptance (review comments 1-3, 5).

>>>Our response

We are very grateful to the Reviewer, who provide all on-spot comments (comments 1, 2, 3, and 5), which guided us in making the revised manuscript better organized and clear than its previous version. We confirm that also comment 4 has been addressed, by providing additional Supplementary Files with the dataset for generating Figure 3 and 4.

Reviewer #2

>>>Reviewer #2

The study entitled “Systematic analysis of SARS-CoV-2 Omicron subvariants’ impact on B and T cell epitopes” presents an extensive analysis of the potential impact of mutations within Omicron subvariants on B- and T-cell epitopes, which is crucial for vaccine and monoclonal antibody development. While this paper makes a significant contribution to the field, it is essential that the information and evidence are well-structured and articulated. Therefore, I believe it is suitable for publication in PLOS ONE.

>>>Our response 

We thank the Reviewer for recognizing the significant contribution that this manuscript conveys. We appreciate the valuable suggestions, which allow us to better articulate our work. In the revised manuscript, we made our best efforts for correcting mistakes and improving the structure as indicated.

>>>Reviewer #2

 Some considerations regarding the following minor points:

 -Change the short title, as it is identical to the main title.

>>>Our response 

We agree with the reviewer that selected editorial metadata entries would benefit from a shorter title, in which case we propose “SARS-CoV-2 Omicron subvariant’s impact on epitopes” to convey a strong message to the readers.

Unfortunately, we did not find an appropriate location for specifying a short title in the Latex template of this Journal. We will inquire with the editorial team if it is possible to add such a shorter title.

>>>Reviewer #2

-The abstract provides relevant introductory information; however, no results are mentioned. Please include the main findings of the paper and any future direction.

>>>Our response 

We agree with the reviewer that the previous version of the abstract could be improved. Here, we clarified our contribution of a workflow and another sentence on the reproducibility of the code, which can be re-run on demand to inform genomic surveillance policy making.

>>>Reviewer #2

-Line 21: “as a candidate to become the most dangerous variant of SARS-CoV-2” appears to demand be a vague statement. Please rephrase for clarity or consider removing it.

>>>Our response

Thank you for noting this. We rephrased the sentence (see Page 2, lines 20-21).

>>>Reviewer #2 

- Check acronyms for consistency. When using an acronym, introduce the full term on first mention, followed by the acronym in brackets, to be used thereafter.

>>>Our response

We carefully revised the whole manuscript; we believe that in the revised version, all the acronyms are properly introduced on their first mention. We apologize for leaving behind cases that are not properly introduced. 

>>>Reviewer #2 

-I am concerned about the timing of retrieving epitope datasets and variant information, which dates back to August 2023, resulting in a lag of over 8 months so far. Could you provide any updates, if possible, at least on the current variants (e.g., JN.1 and its subvariants)?

>>>Our response 

We would like to thank the reviewer for making this highly relevant point. We completely share the reviewer's concern and believe that a short description of an updated analysis would largely benefit the manuscript and make it more interesting for the potential readers.

To this end, we added a new section in the manuscript, named “Analysis reproducibility: focusing on the most recent variants”. Here, we reinforce the message that our analysis workflow is completely repeatable, with openly shared code. As it can be run at any point in time, we did so at the end of May, 2024. The most relevant result is that – for four important new variants (XBB.1.5.70, HK.3, BA.2.86, and JN.1) that – to date – are considered the new variants to be monitored, the percentage of impacted epitopes is even higher than with previous variants. Details are in the revised manuscript on Page 15-16, lines 421-449.

We trust that, with this comment, the reviewer did not imply to completely transform our manuscript with the new analysis, as this would require to regenerate all figures, descriptions, and layout. 

>>>Reviewer #2 

-Considering the length of the materials and methods section, please include only the relevant information in the main text. Add to supplementary.

>>>Our response

We understand the reviewer’s concern. However, we feel that the description of our analysis workflow, given in the Materials and Methods section, provides a clear view of our main contribution in this manuscript: a reproducible analysis pipeline, which can be re-run at any point in time using open data and simple analysis metrics. With this motivation, we would prefer to leave all the relevant text in this section; however, we are open to editing if this is required also on the editorial side.

>>>Reviewer #2 

- In the discussion, provide additional context and background on the role of immunological pressure and proofreading activity on the epitopes impacted by Omicron mutations.

>>>Our response

We would like to thank the reviewer for highlighting this point. In the revised version of the manuscript, we discussed shortly (on Page 18, lines 535-547) the role of immunological pressure on the immune evading ability of Omicron variants, by providing an example that compares it to Delta variant. 

Instead, while we understand the importance of the impact of the proofreading activity on the epitope's regions, in this study, we did not analyze directly such an effect since it is the activity of the Nsp14 exoribonuclease of SARS-CoV-2 which usually repairs errors during replication. Instead, we performed a quantitative and qualitative analysis of the distribution of the T cell and B cell linear epitopes per specific proteins focusing on the Spike. 

>>>Reviewer #2

 - What is the implication of the study's finding on antibody recognition and monoclonal antibody development?

>>>Our response 

We understand the reviewer’s interest in this highly relevant aspect. Please note that, in this study, we did not analyze directly the effects of specific mutations over the antibody recognition and monoclonal antibody. Instead, we gathered the characterizing mutations of Omicron subvariants and quantitatively assessed the epitopes listed in IEDB that have been affected by these mutations. 

On Page 11, lines 312-321, we listed noteworthy Omicron characterizing mutations reported in the literature for decreasing the binding affinity to antibodies or being associated with relative resistance to antibodies.

Moreover, in Figure 5 (discussed on Pages 12-13, lines 336-360), for each amino acid of the Spike protein we count the number of assays that has been performed over the epitopes containing that amino acid and we distinguished between the positive and negative ones. 

A further step is taken in Figure 6 (discussed on Pages 12-14, lines 361-376), where the average values of the response frequency are calculated

---

## [Decision Letter · Decision Letter 1]

15 Jul 2024

Systematic analysis of SARS-CoV-2 Omicron subvariants’ impact on B and T cell epitopes

PONE-D-24-11657R1

Dear Dr. Bernasconi,

We’re pleased to inform you that your manuscript has been judged scientifically suitable for publication and will be formally accepted for publication once it meets all outstanding technical requirements.

Kind regards,

Nagarajan Raju

Academic Editor

PLOS ONE

Additional Editor Comments (optional):

Reviewers' comments:

Reviewer's Responses to Questions

**Comments to the Author**

1. If the authors have adequately addressed your comments raised in a previous round of review and you feel that this manuscript is now acceptable for publication, you may indicate that here to bypass the “Comments to the Author” section, enter your conflict of interest statement in the “Confidential to Editor” section, and submit your "Accept" recommendation.

Reviewer #1: All comments have been addressed

Reviewer #2: All comments have been addressed

Reviewer #3: All comments have been addressed

2. Is the manuscript technically sound, and do the data support the conclusions?

Reviewer #1: Yes

Reviewer #2: Yes

Reviewer #3: Yes

3. Has the statistical analysis been performed appropriately and rigorously? 

Reviewer #1: Yes

Reviewer #2: Yes

Reviewer #3: Yes

4. Have the authors made all data underlying the findings in their manuscript fully available?

Reviewer #1: Yes

Reviewer #2: Yes

Reviewer #3: Yes

5. Is the manuscript presented in an intelligible fashion and written in standard English?

Reviewer #1: Yes

Reviewer #2: Yes

Reviewer #3: Yes

6. Review Comments to the Author

Reviewer #1: (No Response)

Reviewer #2: I appreciate the authors for adequately addressing all comments and I believe the manuscript is suitable for publication in its current form in PLOS ONE.

Reviewer #3: This study presents a comprehensive workflow for analyzing the impact of mutations in SARS-CoV-2 Omicron variants on T cell and B cell epitopes, using data from the Immune Epitope Database, providing valuable insights for genomic surveillance and future epidemic response strategies.

the manuscript has been improved and the comments have been adequately addressed by the authors.

7. PLOS authors have the option to publish the peer review history of their article (what does this mean?). If published, this will include your full peer review and any attached files.

Reviewer #1: No

Reviewer #2: **Yes: **Mohammad Alkhatib

Reviewer #3: No

---

## [Editor Report · Acceptance letter]

17 Jul 2024

PONE-D-24-11657R1 

PLOS ONE

Dear Dr. Bernasconi, 

I'm pleased to inform you that your manuscript has been deemed suitable for publication in PLOS ONE. Congratulations! Your manuscript is now being handed over to our production team.

Kind regards, 

on behalf of

Dr. Nagarajan Raju 

Academic Editor

PLOS ONE